# miR-302 Attenuates Mutant Huntingtin-Induced Cytotoxicity through Restoration of Autophagy and Insulin Sensitivity

**DOI:** 10.3390/ijms22168424

**Published:** 2021-08-05

**Authors:** Ching-Chi Chang, Sing-Hua Tsou, Wei-Jen Chen, Ying-Jui Ho, Hui-Chih Hung, Guang-Yaw Liu, Sandeep Kumar Singh, Hsin-Hua Li, Chih-Li Lin

**Affiliations:** 1Department of Psychiatry, Chung Shan Medical University Hospital, Taichung 402, Taiwan; fmaj7@seed.net.tw; 2School of Medicine, College of Medicine, Chung Shan Medical University, Taichung 402, Taiwan; 3Institute of Medicine, College of Medicine, Chung Shan Medical University, Taichung 402, Taiwan; zinminid@gmail.com (S.-H.T.); liugy@csmu.edu.tw (G.-Y.L.); 4Department of Biomedical Sciences, Chung Shan Medical University, Taichung 402, Taiwan; cwj519@csmu.edu.tw; 5Department of Psychology, Chung Shan Medical University, Taichung 402, Taiwan; yjho@csmu.edu.tw; 6Department of Life Sciences, Institute of Genomics and Bioinformatics, National Chung Hsing University, Taichung 402, Taiwan; hchung@dragon.nchu.edu.tw; 7Indian Scientific Education and Technology Foundation, Lucknow 226002, India; sandeeps.bhu@gmail.com; 8General Education Center, National Taiwan University of Sport, Taichung 404, Taiwan; 9Department of Medical Research, Chung Shan Medical University Hospital, Taichung 402, Taiwan

**Keywords:** autophagy, Huntington’s disease, insulin signaling, mitophagy, miR-302

## Abstract

Huntington’s disease (HD) is an autosomal-dominant brain disorder caused by mutant huntingtin (mHtt). Although the detailed mechanisms remain unclear, the mutational expansion of polyglutamine in mHtt is proposed to induce protein aggregates and neuronal toxicity. Previous studies have shown that the decreased insulin sensitivity is closely related to mHtt-associated impairments in HD patients. However, how mHtt interferes with insulin signaling in neurons is still unknown. In the present study, we used a HD cell model to demonstrate that the miR-302 cluster, an embryonic stem cell-specific polycistronic miRNA, is significantly downregulated in mHtt-Q74-overexpressing neuronal cells. On the contrary, restoration of miR-302 cluster was shown to attenuate mHtt-induced cytotoxicity by improving insulin sensitivity, leading to a reduction of mHtt aggregates through the enhancement of autophagy. In addition, miR-302 also promoted mitophagy and stimulated Sirt1/AMPK-PGC1α pathway thereby preserving mitochondrial function. Taken together, these results highlight the potential role of miR-302 cluster in neuronal cells, and provide a novel mechanism for mHtt-impaired insulin signaling in the pathogenesis of HD.

## 1. Introduction

Huntington’s disease (HD) is a progressive neurological disorder that causes movement, cognitive, and behavioral problems in the central nervous system. Although the incidence of HD is rare, it is the most common dominantly inherited neurodegenerative disease. According to molecular pathology, the cause of HD is due to the mutation of a protein called huntingtin (Htt), which contains a sequence with an expansion of CAG repeats. In most cases, the length of CAG repeats is associated with the onset and progression of HD. When more than 40 CAG repeats are present in Htt alleles, it is inevitable to develop the disease [1]. Expanded CAG repeats in Htt code for an abnormal polyglutamine (polyQ) stretch, and therefore increase the tendency of mutant Htt (mHtt) protein to self-aggregate and deposit in the neuron [2]. Although the detailed mechanisms are still not very clear, these mHtt proteins are found to be toxic to neuronal cells. The accumulation of mHtt has been suggested to interfere with cellular functions including increased oxidative stress, axonal transport defects, impaired synaptic function, decreased autophagy, and therefore eventually leading to neuron death. These pathological changes are mainly found in the basal ganglia and cerebral cortex of HD patients, causing the dysfunction of the cortical-basal ganglia motor control circuit, thereby hindering the control of the patient’s voluntary movement [3].

Although the accumulation of mHtt in the brain is the most important feature of HD, patients also suffer from some type of peripheral symptoms, such as immune disorders, muscle atrophy, osteoporosis and diabetic-like phenotypes [3]. In particular, even though the exact reason is still unknown, many studies have shown a high prevalence of diabetes in HD patients [4]. It is known that the risk of developing diabetes in HD patients is approximately seven times that of the normal control group, suggesting mHtt-induced toxicity may be strongly associated with metabolic disorders [5]. Another recent study also demonstrated that earlier onset patients have been found to have a faster rate of diabetes [6]. The above evidence obviously suggests that HD and diabetes are closely related disorders. In fact, impaired insulin signaling was observed in the development of HD, indicating that a decrease in insulin sensitivity, also known as insulin resistance, is linked to mHtt-associated impairments [7]. Accordantly, our previous study has also reported that compounds increase insulin signaling can enhance protein clearance and therefore reduce the mHtt-induced neurotoxicity in neuronal cells. It seems that the restoration of insulin sensitivity attenuates mHtt-induced oxidative stress and increases autophagy, thereby enhancing the viability of neuronal cells [8]. However, how mHtt interferes with insulin signaling in neurons is still to be determined.

In the central nervous system, the main function of insulin signaling is not only to stimulate the uptake of glucose by neurons. There is clear evidence that insulin signaling plays multiple roles in maintaining neuronal survival and function, and several regulators are known to be involved in the process of affecting neuronal insulin signaling. In this regard, studies have suggested miRNAs may play a role in regulating insulin signaling. miRNA is a short single-stranded non-coding RNA molecule which modulate posttranscriptional expression by targeting specific genes. More specifically, we have previously reported that a miRNA called miR-302 is involved in the pathogenesis of Alzheimer’s disease (AD) [9]. We found that the intracellular levels of miR-302 in the neuronal cells were significantly inhibited in the state of amyloid β (Aβ)-caused neurotoxicity. Conversely, the forced overexpression of miR-302 cluster is sufficient to downregulate expression of phosphatase and tensin homolog (PTEN), which blocks neuronal insulin signaling and therefore aggravates Aβ-induced oxidative stress and apoptosis. Interestingly, the accumulation of misfolded proteins leading to neuronal death is known as a common characteristic in both HD and AD [10]. Since the reduction of miR-302 is involved in the AD-related insulin signaling blockade, it may also contribute to the development of defective insulin signaling in HD pathogenesis. However, we still know very little about the role miR-302 cluster plays in HD. In the present study, we demonstrated that miR-302 was dramatically downregulated in mHtt-overexpressed neuronal cells, and this suppression may be one of the causes of neuronal insulin resistance in the pathogenesis of HD.

## 2. Results

### 2.1. mHtt-Q74 Significantly Reduces miR-302 Cluster Generation in Neuronal SK-N-MC Cells

To establish a cellular model for HD and evaluate the neurotoxic effects of poly-Q mutant huntingtin (mHtt), the human SK-N-MC neuronal cells were transfected with vectors encoding normal Htt (Htt-Q23) or mHtt (Htt-Q74) as previously described [8]. At 48 h following transfection, the cell morphology was observed using a phase-contrast microscope. As shown in Figure 1A, both mock and Htt-Q23-transfected cells did not induce significant toxicity. On the contrary, mHtt-Q74 transfection markedly induced cell death confirming the cytotoxicity. In order to validate whether mHtt is overexpressed, mRNA levels were measured. The results of qPCR showed that mHtt-Q23 and mHtt-Q74 are indeed overexpressed approximately 8 and 13 folds respectively (Figure 1B). The results of MTT assay also showed a significant reduction of cell viability by 45% in mHtt-Q74-overexpressing group (Figure 1C). Next, we investigated the changes in the expression of endogenous miR-302 cluster in neuronal cells. It is known miR-302 cluster is located in the intron between exon 9 and exon 10 of the LARP7 gene, indicating that the intracellular levels of miR-302s generated will be proportional to the splicing rate of these two exons. As a result, as shown in Figure 1D we designed two specific primers on exon 9 and exon 10, respectively, and detected the content of spliced products to estimate the change of miR-302 cluster expression by using qPCR. The results in Figure 1E showed that compared to mock and Htt-Q23 groups, 48 h of mHtt-Q74 overexpression significantly decreased levels of exon 9–10 spliced products, suggesting that overexpression of mHtt-Q74 may result in suppression of miR-302 cluster generation in neuronal cells.

### 2.2. Overexpression of miR-302 Cluster Reduces Apoptosis from mHtt-Induced Neurotoxicity

The above results revealed that the levels of miR-302 cluster are downregulated by overexpression of mHtt-Q74. Therefore, we next investigated whether increasing the expression of miR-302 cluster can reduce cytotoxicity caused by mHtt-Q74. The levels of the vector to overexpress miR-302 cluster was evaluated by qPCR method. Figure 2A showed that miR-302 cluster is indeed overexpressed approximately nine times higher than that in the mock group, suggesting an effective miR-302 cluster overexpression. To test the hypothesis that upregulation of miR-302 may protect cell against mHtt-induced cytotoxicity, cell viability was assessed at 48 h post-transfection. As shown in Figure 2B, the overexpression of miR-302 seems to have no obvious effect on mock- and mHtt-Q23-transfected cells. However, upregulation of miR-302 cluster in mHtt-Q74-overexpressed cells significantly improved cell viability, indicating that miR-302s may indeed play a protective role in mHtt-overexpressing cells. This result was confirmed by the western blot analysis, which showed that mHtt-Q74 increased the cleaved forms of both caspase 3 and PARP, whereas upregulation of miR-302 markedly attenuated mHtt-induced apoptosis (Figure 2C). The apoptosis was also confirmed by the results of the DAPI nuclear staining assay. As shown in Figure 2D, fragmented nuclear morphology was significantly enhanced in the mHtt-Q74-overexpressing group. Accordingly, mHtt-Q74-induced apoptosis was confirmed again by terminal deoxynucleotide transferase dUTP nick-end labeling (TUNEL) assay, which demonstrates that overexpression of mHtt-Q74 significantly decreased the number of apoptotic TUNEL-positive cells. Conversely, when miR-302 was co-overexpressed at the same time, the rate of occurrence of TUNEL-positive cells can be markedly reduced (Figure 2E). These results collectively demonstrated that mHtt-induced apoptosis could be prevented by upregulation of miR-302 cluster.

### 2.3. Upregulation of miR-302 Cluster Facilitates Mitochondrial Autophagy in mHtt-Overexpressing Cells

A significant decline in autophagy activity has been described in HD cells [11]. Therefore, we next investigated whether miR-302 affects the activity of autophagy in mHtt-transfected neuronal cells. The results of acridine orange (AO) staining were shown in Figure 3A, demonstrating that compared to the mock group, mHtt-Q74 transfection significantly decreased the average number of acidic vesicular organelles (AVOs) per cell from 9.5 ± 2.3 to 1.0 ± 1.2. However, co-transfection with miR-302 cluster effectively restored mHtt-mediated AVOs reduction, indicating the upregulation of miR-302 cluster may induce autophagy (Figure 3B). The results of immunoblotting also provided further evidence that mHtt-Q74 transfection markedly reduces LC3-II expression, whereas forced overexpression of miR-302 reversed the level of LC3-II (Figure 3C). In addition to western blotting for processed LC3, levels of the p62 and ATG5, two autophagy related proteins were also restored by miR-302, showing that the overexpression of miR-302 can indeed effectively reverse the autophagy suppressed by mHtt-Q74. The levels of LC3-II protein were also increased by autophagy-modifying compounds rapamycin (Rapa) and chloroquine (CQ) co-treatment, suggesting the involvement of autophagic flux by miR-302 cluster. Previous studies have indicated that mitochondrial dysfunction can be linked to the pathology of HD. In particular, abnormalities at mitochondrial autophagy, a process also called mitophagy, has been proposed to contribute to disease progression [12]. Therefore, we next investigated whether miR-302 improves mitophagy in mHtt-transfected neuronal cells. As shown in Figure 3D, the colocalization of autophagy marker LC3 and mitochondria marker was obviously reduced in mHtt-Q74-transfected cells, suggesting mitophagy was greatly suppressed. By contrast, mitochondria co-localized with LC3 was markedly increased by overexpression of miR-302 cluster, indicating that upregulation of miR-302s may facilitate mitophagy in mHtt-overexpressing cells.

### 2.4. miR-302 Cluster Upregulates Sirt1/AMPK-PGC1α Pathway through the Restoration of Insulin Signaling

Previous studies have indicated that the impaired insulin signaling may be linked to the mHtt-mediated neurotoxicity [13]. To determine whether miR-302 confers its protective effect by affecting insulin signaling activity, we performed western blotting analysis to detect the phosphorylation levels of insulin receptor substrate-1 (IRS-1) at residue Ser^307^, which is a hallmark that blocks downstream insulin signaling. As shown in Figure 4A, mHtt-Q74 overexpression increased Ser^307^ IRS-1 phosphorylation compared to the mock-treated groups. Accordingly, the phosphorylation of downstream target Ser^473^ Akt also decreased in the Htt-Q74-overexpressing group, implying neuronal insulin signaling is indeed inhibited by mHtt-Q74. By contrast, co-transfected with miR-302 cluster dramatically reduced IRS-1 Ser^307^ phosphorylation and a subsequent restoration in Akt Ser^473^ phosphorylation, suggesting mHtt-induced insulin signaling blockade can be restored by miR-302 cluster. In order to confirm the role of miR-302 cluster on insulin signaling, the specific PI3K inhibitor LY294002 was treated as a negative control. As demonstrated in Figure 4B, the MTT assay results showed that miR-302 cluster effectively alleviated mHtt-Q74-induced cytotoxicity, whereas co-treatment with LY294002 blocked this protective effect. This indicated that miR-302s-mediated insulin signaling is dependent on insulin signaling transduction. As mentioned earlier, reduced mitophagy is linked to neuronal dysfunction and death of HD [14]. We next examined the protein changes related to mitophagy and mitochondrial turnover. As shown in Figure 4C, western blotting results demonstrated that in mHtt-Q74-overexpressing cells, being co-transfected with miR-302 markedly increases AMPK Thr^172^ phosphorylation and Sirt1, which have been known to stimulate the expression of ATGs with subsequent induction of mitophagy [15]. In accordance with this observation, the AMPK/Sirt1 downstream mitochondrial biogenesis factor peroxisome proliferator-activated receptor gamma coactivator 1α (PGC1α) was also upregulated. However, exposure to LY294002 attenuated most of the effects caused by miR-302 cluster. The results of JC-1 staining also demonstrated that overexpression miR-302 cluster preserves mitochondrial function against mHtt-induced mitochondrial dysfunction dependent on insulin signaling (Figure 4D). Collectively, our results indicated miR-302 upregulates Sirt1/AMPK-PGC1α pathway in maintaining mitochondrial function through the restoration of insulin signaling.

### 2.5. Upregulation of miR-302 Cluster Reduces the Number and Size of mHtt Aggregates in mHtt-Overexpressing Cells

In the pathology of HD, the size and number of mHtt intracellular aggregates are directly associated with neuronal cell death. Therefore, we analyzed the formation of aggregate-like structures by immunocytochemistry. As shown in Figure 5A, immunostaining of Htt showed that overexpression of mHtt-Q74 markedly causes an increased number of aggregates, whereas the upregulation of miR-302 cluster displays the opposite effect in aggregate formation. Interestingly, the exposure of LY294002 removed the inhibitory effect of miR-302 cluster on mHtt aggregates, showing that the induction of autophagy seems to promote the clearance and degradation of mHtt. In order to measure the number and size of mHtt aggregates, the images were further analyzed by automated high-content analysis (HCA). As shown in Figure 5B, neuronal cells transfected with mHtt-Q74 significantly increased in the number of Htt-puncta per cell compared to the mock-treated group. However, co-overexpression of miR-302 cluster greatly decreased average number of mHtt aggregates from 47 to 9 puncta per cell, and this inhibition could also be reversed by the addition of LY294002. Furthermore, the measurements of aggregate size distribution also demonstrated that mHtt aggregates shifted towards smaller sizes by upregulation of miR-302, whereas LY294002 markedly reversed this effect (Figure 5C). Taken together, these results imply that upregulation of miR-302 cluster effectively reduces the number and size of mHtt aggregates in mHtt-overexpressing cells.

## 3. Discussion

So far, HD is still an incurable neurodegenerative disorder. Since the cause of HD is the mutation of Htt gene, it is difficult to develop effective strategies for complete success in therapy for HD. However, although patients are born with mutated genes, neurodegeneration is not found in infants and young children. In most cases, symptoms appear after early adulthood, implying that the mutation of Htt is only one of the factors that induce HD symptoms, and there are other factors that may also contribute to the pathogenesis. As such, autophagy is becoming recognized as a cellular response to protect against neurodegeneration [16]. In particular, for larger protein aggregates such as mHtt, adequately activated autophagy seems to be more effective in cell self-clearance. In our current study, we found that when cells overexpress mHtt-Q74, the intracellular levels of miR-302 cluster decrease significantly. Conversely, increasing the reduced expression of miR-302 cluster in mHtt-overexpressing cells effectively inhibited mHtt-induced neurotoxicity, indicating that miR-302 is likely to be involved the mechanism of attenuating mHtt-induced cytotoxicity. We also demonstrated that the induction of miR-302 may reduce the number and size of mHtt aggregates by upregulating the activity of autophagy, and this process is closely related to neuronal insulin signaling. In fact, previous research has reported that autophagy induced by insulin signaling may be an effective approach for mHtt aggregate clearance [17]. Accordingly, as we previously discovered, we found that activation of miR-302s in neuronal cells can enhance cellular survival under stress by improving autophagic activity [9]. As such, we suggest that the upregulation of miR-302 may be a potential strategy to reduce mHtt neurotoxicity.

Autophagy is known as a cellular survival pathway in the elimination of pathogenic intracellular protein aggregates. Particularly, degradation of mitochondria through the autophagy, also termed as mitophagy, is crucial to maintain mitochondrial turnover and biogenesis in neurons. In fact, a decline in mitochondrial quality and dynamics has been linked to neuronal aging-associated changes of neurodegenerative diseases. Regarding this, the miR-302 cluster was first discovered to play an important role in maintaining stemness of human embryonic stem cells [18]. It is known that aging is associated with loss of cellular stemness resulting in decreased capacity to repair tissues [19]. On the contrary, miR-302 has been confirmed to have the ability to enhance stemness for most cell types [20,21,22], suggesting the restoration of miR-302 expression may promote survival or regeneration of damaged neuron cells, and consequently delay the onset or progression of HD. It has been seen that the increase of stemness can prolong neuronal survival by upregulating neurotrophic factors in damaged neuronal cells [23], implying that promoting miR-302 cluster in the brain is a feasible strategy to reduce, delay or prevent symptoms caused by mHtt. For example, using specific compounds to induce endogenous miR-302s expression, or directly increasing the brain miR-302 content through lipid-based delivery methods, both of which may become future directions for the treatment of HD.

On the other hand, defective insulin signaling has long been considered to be involved in the pathophysiology of HD. It has been estimated that the risk of type 2 diabetes in HD patients is about seven times higher than that in the control group [5], suggesting insulin resistance might contribute to promoting the HD pathologies. Notably, a large body of evidence has been accumulated to support the idea that insulin signaling in the brain is essential for regulating vital neuronal and glial features [24]. Neuronal insulin signaling plays important roles in neurite growth, synaptic plasticity, neuronal survival, and mitochondrial function. In the process of mHtt-caused neurotoxicity, the above-mentioned functions can be strongly affected, which in turn interferes with the normal functions of neurons and eventually leads to apoptosis. In accordance with our results, we observed that overexpression of mHtt in neuronal cells markedly blocks insulin signaling and results in a condition similar to insulin resistance. In fact, our previous study has indicated that miR-302 is associated with the maintenance of neuronal insulin sensitivity [9]. Accordingly, previous studies also demonstrated that increased levels of glucagon-like peptide-1 (GLP-1), an inducer of insulin secretion, can lead to activation of Akt signaling and higher brain LC3-II levels, thus promoting autophagy [25]. In this study, the overexpression of mHtt leads to a significantly decreased endogenous level of miR-302 cluster, showing that mHtt may induce neuronal insulin signaling inhibition by suppressing miR-302. Interestingly, insulin/Akt signaling has been identified to modulate fundamental stem cell activities through retention of viability and expression of pluripotency-associated characteristics [26]. This also highlights that regulation of miR-302s on insulin signaling is the key mechanism to establish the property of neural stemness that may facilitate neuronal repair and regeneration against mHtt-induced neurodegeneration. However, until now, no study has been undertaken regarding the change of miR-302 cluster in the brains of HD patients, and further study is required to investigate the interaction between miR-302 and mHtt in vivo. Taken together, our observations confirm that overexpression of mHtt in neuronal cells significantly suppresses the endogenous level of miR-302, which provides a novel mechanism for mHtt-impaired insulin signaling and contributes to the development of new approaches for treating HD in future.

## 4. Materials and Methods

### 4.1. Materials

Chemicals such as 3-(4,5-dimethylthiazol-2-yl)-2,5-diphenyltetrazolium bromide (MTT), 4′,6-diamidino-2-phenylindole (DAPI), Hoechst 33342, acridine orange (AO), JC-1, chloroquine, rapamycin and LY294002 were purchased from Sigma (München, Germany). The Mitotracker Green FM was supplied by Thermo Fisher Scientific (Waltham, MA, USA). We purchased pHM6-Q23 Htt (Plasmid# 40263) and pHM6-Q74 Htt (Plasmid# 40264) from Addgene (Cambridge, MA, USA). Antibodies against caspase 3, and poly(ADP-ribose) polymerase (PARP), Akt, p-Akt, AMPK, p-AMPK, p62, PCG1α were obtained from Santa Cruz Biotechnology (Santa Cruz, CA, USA), and antibody against Sirt1 was purchased from GeneTex (Irvine, CA, USA). The IRS-1, p-IRS-1, Atg5 antibodies was obtained from Cell Signaling Technology (Danvers, MA, USA), and β-actin and LC3 antibodies were purchased from Novus Biologicals (Littleton, CO, USA). Primary antibodies were used at a dilution of 1:200 (ICC) or 1:1000 (WB) in 0.1% Tween 20, and secondary antibodies were used at 1:1000–1:5000 dilutions depending on the individual sample conditions.

### 4.2. Cell Culture, Transfection and Viability Assay

Human SK-N-MC neuroblastoma cells were obtained from the American Type Culture Collection (Bethesda, MD, USA). Cells were cultured in Minimal Eagle’s medium (Gibco, Carlsbad, CA, USA) supplemented with 10% fetal calf serum, antibiotics (100 units/mL penicillin, 100 µg/mL streptomycin), 2 mM L-glutamine, and cultivated at 37 °C in a humidified atmosphere containing. 5% CO_2_. Lipofectamine 2000 transfection reagent (Thermo Fisher Scientific, Waltham, MA, USA) was used for transient mutant huntingtin transfections following manufacturer’s protocol. For overexpressing miR-302 cluster, a miR-302s vector was modified from Clontech’s pTet-On-tTS-miR302 cluster plasmid (miR-302b*-b-c*-c-a*-a-d) as previously reported [27]. Then, the cells were transfected with the mir302 cluster vector to generate miR-302s-overexpressed cells, and validated by real-time qPCR. Transfection was performed using 4 µL lipofectamine 2000 and 2 µg vector for 2 × 10^5^ cells. In MTT assays, cells were treated with tetrazolium salt for 30 min at room temperature. Then, the formazan product was measured spectrophotometrically at 550 nm. Viability was calculated as percent of control cells treated with vehicle alone.

### 4.3. mRNA Expression Analysis by Reverse-Transcription Quantitative PCR (qPCR)

Extraction of total RNA from cultured cells was performed by using RNeasy Kit (Qiagen, Germantown, MD, USA). After the isolation, cDNA was reverse transcribed from total RNA using TProfessional Thermocycler Biometra (Göttingen, Germany) following the manufacturer’s recommendations (high-capacity cDNA reverse transcription kit, Applied Biosystems, Foster City, CA, USA). For quantitative real-time PCR, reactions were performed using Power SYBR Green PCR master mix an ABI 7300 Real Time PCR System (Applied Biosystems, Foster City, CA, USA). qPCR parameters were formed as previously described [8], and the primer pairs used for detecting spliced exon 9 and exon 10 of LARP7 were forward 5′-AGAGT GCTAT CAAAG AGCGA ATGG-3′ and reverse 5′-TGAGG TACTC CACTG TCTGT TTCC-3′. The following primer pairs were also used: forward 5′-CACTG CTGGA CAGAT TCCGA-3′ and reverse 5′- ACTGG TATGA TGTGG TATCA CC -3′ for Htt exon 1 coding sequence; forward 5′-TGAAT CCAAT TTACT TCTCC A-3′ and reverse 5′- TCCTT TAACC TGTAA CAAGC-3′ for pri-miR-302. Quantitative gene expression data were normalized to the expression levels of glyceraldehyde 3-phosphate dehydrogenase (GAPDH) using primer pairs with forward 5′-TGGTAT CGTGG AAGGA CTCAT GAC-3′ and reverse 5′-ATGCC AGTGA GCTTC CCGTT CAGC-3′. All data were performed in triplicate, and quantitatively analyzed on an ABI Prism 7300 Sequence Detection System (SDS) with SDS software, version 1.2.3 (Applied Biosystems, Foster City, CA, USA) using the ΔΔCt method.

### 4.4. Western Blot Analysis

For western blotting, cells were lysed in Gold Lysis Buffer (50 mM Tris-HCl, pH 8.0, 5 mM ethylenediaminetetraacetic acid, 150 mM NaCl, 0.5% Nonidet P-40, 0.5 mM phenylmethylsulfonyl fluoride, and 0.5 mM dithiothreitol). Equivalent amounts of protein lysates were electrophoresed by 8–10% SDS-polyacrylamide gels. The gels were then electroblotted onto PVDF membranes (Millipore, Bedford, MA, USA) and incubated with primary antibodies followed by secondary antibodies conjugated with horseradish peroxidase. The relevant protein was visualized by enhanced chemiluminescence assay and detected by the AI600 imager system (GE Healthcare, Chicago, IL, USA).

### 4.5. DAPI Staining on Nuclei

Cells were fixed in 4% paraformaldehyde for 24 h at 4 °C, and then incubated with DAPI stain solution (1 ng/mL in Mcllvaine’s buffer, pH 7.0) in dark for 30 min at room temperature. Images were taken using a fluorescent microscope (DP80/BX53, Olympus, Tokyo, Japan), and the percentages of apoptotic fragmented nuclei for each sample were calculated in five randomly selected fields for each group.

### 4.6. TdT-Mediated dUTP-X Nick End Labeling (TUNEL) Assay

TUNEL assay was performed with a detection kit following the manufacturer’s instructions (Abcam, Cambridge, MA, USA). Briefly, the cells were washed with PBS and fixed with 4% paraformaldehyde for 15 min at room temperature, and then the incubated in permeabilization solution (buffered 0.1% Triton X-100). Fixed cells were incubated with the TUNEL reaction mixture in an incubator at 37 °C for 1 h, and then washed with PBS and counterstained with DAPI for 5 min in order to visualize the nuclei. Subsequently, images were examined by a fluorescence microscope (DP80/BX53, Olympus, Tokyo, Japan), and images of five random and non-overlapping fields were calculated.

### 4.7. Immunocytochemistry and Acridine Orange (AO) Staining

Cells were fixed in 4% paraformaldehyde for 24 h at 4 °C, permeabilized in 0.1% Triton X-100 (Sigma-Aldrich, München, Germany) for 15 min at 4 °C, and incubated in blocking solution (10% FBS in PBS) for 24 h at 4 °C. Fixed cells were incubated with primary antibodies for 2–6 h at room temperature, and then incubated with Rhodamine- or FITC-labeled second antibody (Santa Cruz Biotechnology, Santa Cruz, CA, USA) corresponding to primary antibody. For acridine orange (AO) staining, cells were stained with 1 μg/mL AO for 15 min at room temperature and washed twice with medium. Fluorescence images were acquired by using a fluorescence microscope (DP80/BX53, Olympus, Tokyo, Japan), and the number of acidic vesicular organelles (AVOs) per cell was counted in five randomly selected fields for each group.

### 4.8. Analysis of Mitochondrial Membrane Potential by JC-1 Staining

Cells were stained with 1 µM JC-1 for 30 min at 37 °C in dark, and then washed twice with PBS. Fluorescence images were taken using by using an inverted fluorescence microscope (DP72/CKX41, Olympus, Tokyo, Japan). Red JC-1 fluorescence indicates healthy mitochondria, and green JC-1 fluorescence represented loss of mitochondrial membrane potential. The quantitative data analysis of red/green ratio were be performed with Image J (version 1.8.0, National Institutes of Health, Bethesda, MD, USA).

### 4.9. High-Content Fluorescence Microscopy

Cells were seeded in 24-well black imaging plates at a density of 1 × 10^6^ cells per well and allowed to adhere. The adapted cells were then treated with the indicated conditions, and fixed with 4% paraformaldehyde at 4 °C supplemented with 1 µg/mL Hoechst 33,342 in PBS overnight. Subsequently, the plates were analyzed by automated microscopy. Images were acquired by an ImageXpress Micro Confocal High-Content imaging system (Molecular Devices, Sunnyvale, CA, USA). Parameters of rhodamine-positive particles were automatically determined; this was performed by using the custom module editor within the MetaXpress software (Molecular Devices, Sunnyvale, CA, USA). At least eight view fields were acquired per well, and each condition involved at least ran in triplicate assessment.

### 4.10. Statistical Analysis

All data in this study were presented as the means ± SEM, followed by the two-tailed Student’s t test for quantitative variables between two groups, or one-way analysis of variance (ANOVA) followed by post hoc Dunnett’s multiple comparison test for three or more groups. Statistical analysis data analysis was performed by Graph-Pad Prism 8.0, and *p*-values below 0.05 were considered statistically significant.

## Figures and Tables

**Figure 1 ijms-22-08424-f001:**
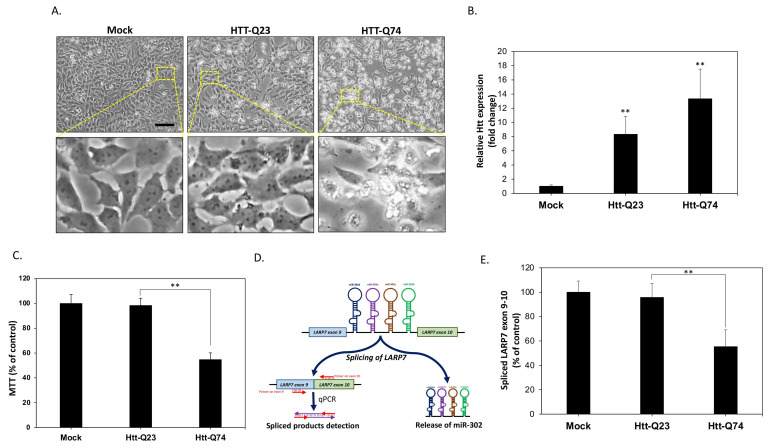
The generation of endogenous miR-302 is downregulated in mHtt-Q74-overexpressing neuronal cells. (**A**) Human SK-N-MC neuronal cells are transfected with the empty vector (mock), normal polyQ Htt (Htt-Q23), or polyQ-expanded mHtt (mHtt-Q74). Cell images are taken after 48 h of transfection, showing that no obvious morphological change of cytotoxicity is observed in both mock and Htt-Q23-transfected cells. However, cytotoxic features such as cell shrinkage are markedly increased in cells overexpressing mHtt-Q74. (**B**) The qPCR results show significantly increased mHtt mRNA levels, suggesting effective overexpression. (**C**) The MTT results indicate that after 48 h of transfection, cell viability is significantly reduced by approximately 55% in the mHtt-Q74-overexpressing group. (**D**) A proposed detection scheme demonstrates that the amount of intracellular miR-302 cluster can be obtained by indirectly measuring LARP7 exon 9–10 spliced products through a designed qPCR method. (**E**) The results of qPCR reveal that overexpression of mHtt-Q74 causes a significant decrease in exon 9–10 spliced products of LARP, suggesting the levels of miR-302 cluster are also suppressed. All values are presented as mean ± SEM. Significant difference was determined by using the multiple comparisons of Dunnett’s posthoc test for ** *p* < 0.01. Scale bar represents 50 μm.

**Figure 2 ijms-22-08424-f002:**
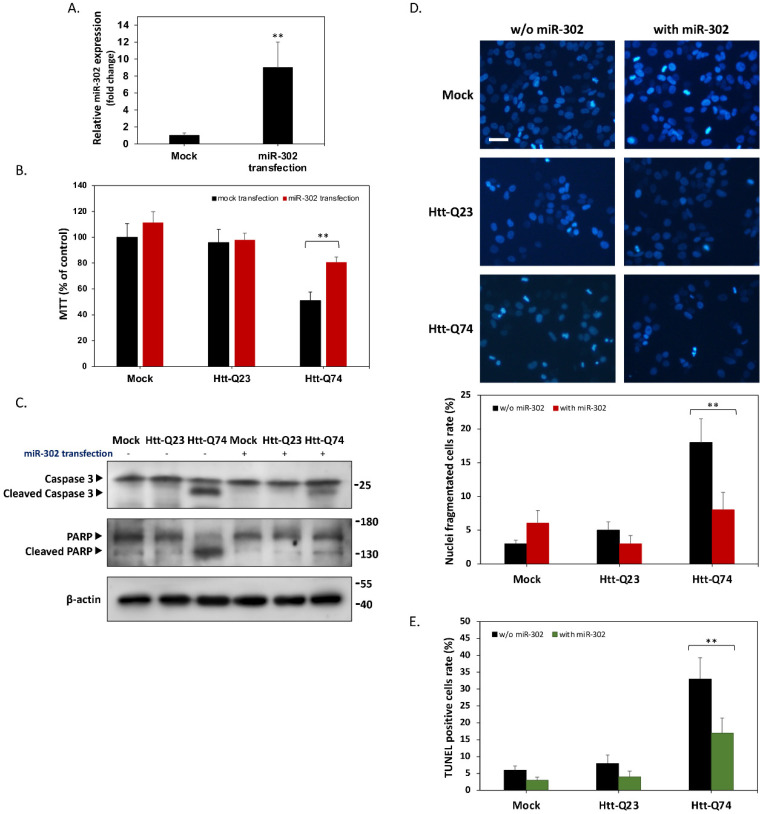
Upregulation of miR-302 cluster protects neuronal cells from Htt-Q74-induced apoptosis. (**A**) SK-N-MC neuronal cells are transfected with miR-302 overexpressing vector for 48 h. The qPCR results indicate the successful miR-302 overexpression in transfected cells. (**B**) Cell viability is measured by MTT assay 48 h post-transfection. The result indicates that overexpression of miR-302 cluster significantly improves cell viability in mHtt-Q74-overexpressed cells. (**C**) Western blot analysis demonstrates that upregulation of miR-302 cluster inhibits cleavage of caspase 3 and PARP, two apoptotic markers involved in mHtt-induced neurotoxicity. (**D**) Nuclear fragmentation examined by DAPI cytochemistry staining. mHtt-Q74-induced apoptotic nuclear fragmentation is markedly decreased by co-transfection of miR-302. (**E**) Cell are stained with the terminal deoxynucleotide transferase dUTP nick-end labeling (TUNEL) method, and apoptotic cells are counted. All values are presented as mean ± SEM. Significant difference was determined by using the multiple comparisons of Dunnett’s posthoc test for ** *p* < 0.01. Scale bar represents 20 μm.

**Figure 3 ijms-22-08424-f003:**
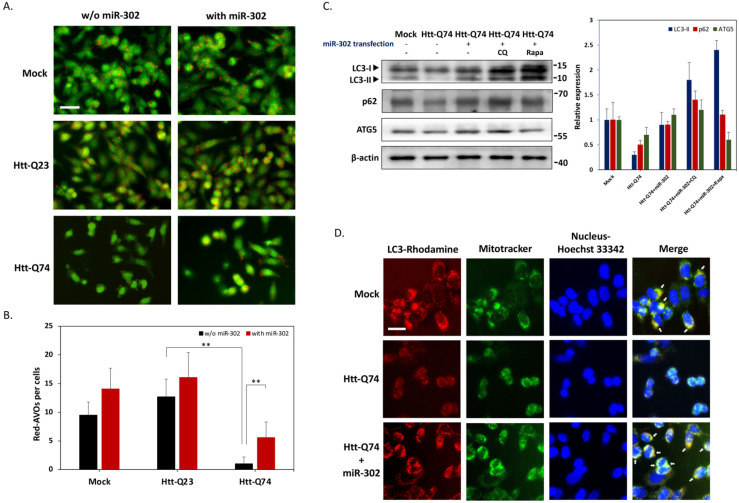
(**A**) Fluorescence microscopy images of acridine orange (AO) staining cells, and autophagy induction was indicated by increased in number of red cytoplasmic dots. Scale bar represents 20 μm. (**B**) Bar graphs show the average number of AVOs per cell calculated from five images of each treatment. (**C**) Western blot analysis for autophagy-related proteins including ATG5, p62, and. LC3B. Specific protein bands were quantified and normalized to β-actin, and shown as fold change. CQ: chloroquine (50 μM); Rapa: rapamycin (20 nM). (**D**) Immunofluorescence staining was conducted for LC3 (red), Mitotracker (green) and nucleus (blue). Arrowheads indicate overlap between LC3 and Mitotracker (yellow), suggesting these two proteins could be co-localized. Scale bar represents 5 μm. All values are presented as mean ± SEM. Significant difference was determined by using the multiple comparisons of Dunnett’s posthoc test for ** *p* < 0.01.

**Figure 4 ijms-22-08424-f004:**
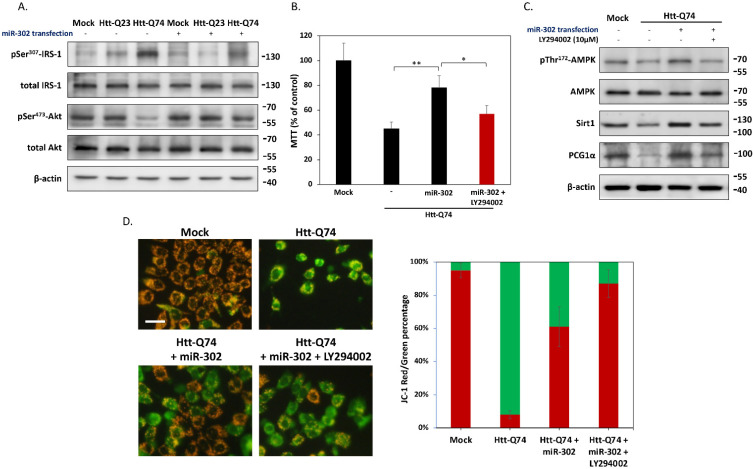
miR-302 upregulates Sirt1/AMPK-PGC1α pathway and preserves mitochondrial function. (**A**) The protein levels of IRS-1 and downstream molecules Akt by using western blotting. (**B**) MTT assays show the LY294002 co-treatment inhibits the protective effect of miR-302 in mHtt-Q74-overexpressing cells. (**C**) Western blots show that miR-302 upregulates Sirt1, AMPK and PGC1α, three proteins positively associated with mitochondrial biogenesis and mitophagy. On the contrary, co-treatment with 20 μM LY294002 markedly blocks the effects caused by miR-302, suggesting the protection by miR-302 is dependent on insulin signaling. (**D**) Representative images of mitochondrial membrane potential are demonstrated by JC-1 staining, and cells observed with green and red fluorescence are also quantified in percentages. The increase of JC-red shows that miR-302 can protect against mHtt-induced mitochondrial dysfunction (JC-green). However, LY294002 prevents the effects of miR-302, suggesting that insulin signaling is essential for the protection of miR-302. All values are presented as mean ± SEM. Significant difference was determined by using the multiple comparisons of Dunnett’s posthoc test for * *p* < 0.05 and ** *p* < 0.01. Scale bar represents 10 μm.

**Figure 5 ijms-22-08424-f005:**
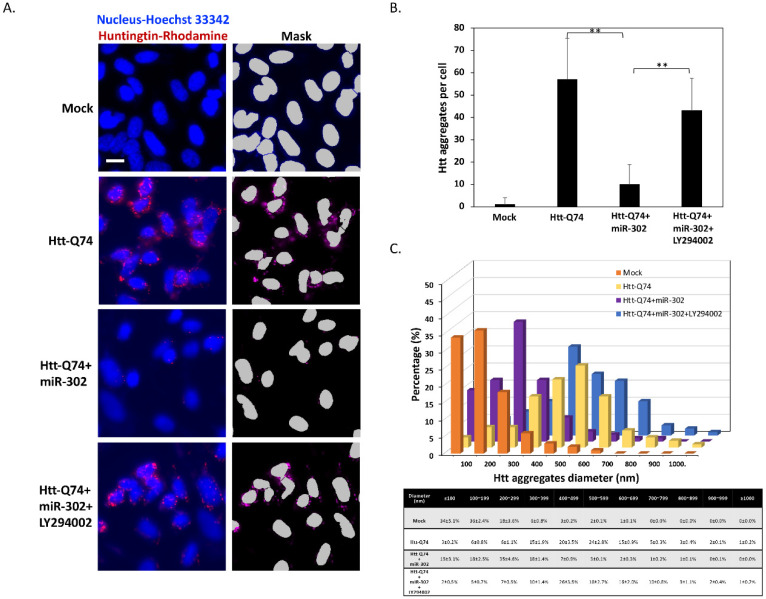
miR-302 cluster reduces the number and size of mHtt aggregates. (**A**) Immunofluorescence staining of Htt shows that 48 h of Htt-Q74 overexpression promotes the formation of Htt aggregates. However, the aggregation is markedly decreased by upregulation of miR-302 cluster. (**B**) The results of the automated high-content analysis indicate that miR-302 significantly reduces average number of mHtt aggregates in mHtt-overexpressing cells. (**C**) the measurements of aggregate demonstrate that mHtt aggregates shifted towards smaller sizes by upregulation of miR-302, whereas co-treated with LY294002 reverses this reduction. All values are presented as mean ± SEM. Significant difference was determined by using the multiple comparisons of Dunnett’s posthoc test for ** *p* < 0.01. Scale bar represents 5 μm.

## Data Availability

All the data are shown in the main manuscript.

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
