# Peer review of "miR-302 Attenuates Mutant Huntingtin-Induced Cytotoxicity through Restoration of Autophagy and Insulin Sensitivity"

_ijms, 2021, doi:10.3390/ijms22168424_

Round 1

Reviewer 1 Report

The authors in accordance with the instructions: "Tn order to ensure the integrity and scientific validity of blots (including but not limited to Western blots) and gel data reporting, original, uncropped and unadjusted images should be uploaded as Supporting Information files at time of initial submission.. .........." must modify the Western Blots present in the figures and must add the indications of the corresponding MWs.the manuscript is clear enough the methodologies are correct and the results discussed, the recent bibliography should be reported.

Author Response

#Reviewer 1

The authors in accordance with the instructions: "Tn order to ensure the integrity and scientific validity of blots (including but not limited to Western blots) and gel data reporting, original, uncropped and unadjusted images should be uploaded as Supporting Information files at time of initial submission. .........." must modify the Western Blots present in the figures and must add the indications of the corresponding MWs. the manuscript is clear enough the methodologies are correct and the results discussed, the recent bibliography should be reported

Answer: Thanks for reviewer’s comments. By your suggestion, we have added the indications of the corresponding MWs for each WB blots. In addition, we have also revised the manuscript and updated some recent references. Hope these changes can improve our research and meet the requirements. Anyway, sincerely thank you for your valuable suggestions.

Reviewer 2 Report

In the study by Chang C-C et al titled “miR-302 attenuates mutant huntingtin-induced cytotoxicity through restoration of autophagy and insulin sensitivity” the authors investigate the link between miR-302, an embryonic stem cells-specific miRNA, with mutant huntingtin (mHtt).

Their main results has been: 1) miR-302 is significantly downregulated in mHtt-Q74-overexpressing neuronal cells; 2) Restoration of miR-302 was shown to attenuate mHtt-induced cytotoxicity by improving insulin sensitivity, and reducing mHtt aggregates through the enhancement of autophagy; 3) miR-302 also promoted mitophagy and stimulated Sirt1/AMPK-PGC1α pathway thereby preserving mitochondrial function.

The study shows a novel mechanism for mHtt-impaired insulin signaling in the pathogenesis of Huntington Disease.

Results have well been explained and are convincing.

Main remarks

  1. The authors should explain the shown images from Fig 1A. Are there difference in cellular morphology in the three conditions?
  2. In order to verify accumulation of p62 protein the authors should use rapamycin as autophagy activator and include in Fig 3C.

Author Response

#Reviewer 2

In the study by Chang C-C et al titled “miR-302 attenuates mutant huntingtin-induced cytotoxicity through restoration of autophagy and insulin sensitivity” the authors investigate the link between miR-302, an embryonic stem cells-specific miRNA, with mutant huntingtin (mHtt).

Their main results has been: 1) miR-302 is significantly downregulated in mHtt-Q74-overexpressing neuronal cells; 2) Restoration of miR-302 was shown to attenuate mHtt-induced cytotoxicity by improving insulin sensitivity, and reducing mHtt aggregates through the enhancement of autophagy; 3) miR-302 also promoted mitophagy and stimulated Sirt1/AMPK-PGC1α pathway thereby preserving mitochondrial function.

The study shows a novel mechanism for mHtt-impaired insulin signaling in the pathogenesis of Huntington Disease.

Results have well been explained and are convincing.

Main remarks

The authors should explain the shown images from Fig 1A. Are there difference in cellular morphology in the three conditions?

Answer: The main purpose of Fig. 1A is to show that compared with mock and Htt-Q23 groups, 48 h overexpressing Htt-Q74 does have a tendency to induce cytotoxicity. We can find that cell shrinkage occurs in neuronal cells with Q74 overexpression, and this phenomenon implies that it may be due to apoptosis. Therefore, in Fig. 2C/D/E we analyzed the markers related to apoptosis and confirmed that Q74 is indeed undergoing apoptosis, and this phenomenon is consistent with the known HD pathology. In this regard, in this revision, we have added enlarged images in Fig. 1A to clearly show the changes in cellular morphology of the three conditions.

In order to verify accumulation of p62 protein the authors should use rapamycin as autophagy activator and include in Fig 3C.

Answer: By reviewer’s suggestion, we have added the results of rapamycin or Chloroquine in Fig. 3C. In this result we found that the treatment of autophagy inducer Rapamycin significantly increased the expression of LC3-II, confirming autophagy has indeed been induced. However, the levels of p62 did not change significantly, or even decreased a little. We think this means that the effect of lysosome degradation protein is smoother after autophagy is induced, so the total amount of p62 is slightly reduced.

Reviewer 3 Report

In this manuscript, the authors studied the protective role of miR302 on mHtt induced neuronal toxicity. In addition, the authors tried to provide the link between insulin sensitivity and Huntington's disease pathology. The theme and linking point are very interesting and would be helpful insights to understand neurodegenerative disease and to approach the whole system, not just in the brain event. Authors already studied and confirmed the role of miR302 on Alzheimer's disease. In the present study, the authors made it widen in Huntington's disease.

Applying several assays used and provided the results, authors were tried to support the conclusion. However, this study suggests too limited data to combine the points suggested by the authors. There are several things to be crucially provided.

  1. Figure 1A showed that three kinds of exogenous genes were transformed to overexpress mHtt-Q74 in neuronal cells. The three plasmids were empty vector, Q23, and Q74. As described in "Materials and methods," the used plasmid DNA were GFP tagged construct, which can confirm the genes successfully transfected and expressed in cells. In this Figure, the GFP signals are needed to provide along with the bright field images. With the present images, it cannot validate whether Q74 was overexpressed in SK-N-MC. Only an increase of cell death in the cells transfected Q74 did not prove the overexpression of Q74.
  2. The construct of miR302 used in this study is needed in detail. According to "Materials and methods," the pLVX-AcGFP vector was modified for miR302 overexpression, and Figure 2A presented dsRed-RFP for confirming miR302 expression. It might that dsRed was fusioned to miR302 because Q74 was tagged with GFP.
  3. Nuclear fragmentation was assessed just by DAPI staining, and it was quantified…….. I do not think it is an acceptable way. I recommend TUNEL staining for nuclear fragmentation.
  4. There are needed to explain fluorescence images in Figure 3A, Figure 3D, Figure 4D, and Figure 5A. Q74 was tagged with GFP, and miR302 was tagged with dsRed. How could each fluorescence detect except tagging fluorescence proteins in Q74 and miR302 overexpressed neuronal cells? Another construct of Q74 or miR302 is used for Figure 3A, Figure 3D, Figure 4D, and Figure 5A? If it was, how could determine to overexpress Q74 or miR303 in the transfected cells?
  5. Figure 5A showed the stained Htt with rhodamine to evaluate Htt aggregates. Why? Q74 was already tagged with GFP, and the GFP signals were shown as puncta in the nucleus of Q74 overexpressed cells. In addition, if the authors would suggest miR302 might work on Q74 overexpressing cells, co-localization of Q74 and miR302 in the same neuronal cells was displayed at least (/primarily).
  6. For detecting or quantifying the level of miR302, TaqMan assay was general, and there are commercial probes for miR302. Is there any specific reason for measuring the LARP7 exon 9-10 spliced product?

Author Response

#Reviewer 3

In this manuscript, the authors studied the protective role of miR302 on mHtt induced neuronal toxicity. In addition, the authors tried to provide the link between insulin sensitivity and Huntington's disease pathology. The theme and linking point are very interesting and would be helpful insights to understand neurodegenerative disease and to approach the whole system, not just in the brain event. Authors already studied and confirmed the role of miR302 on Alzheimer's disease. In the present study, the authors made it widen in Huntington's disease.

Applying several assays used and provided the results, authors were tried to support the conclusion. However, this study suggests too limited data to combine the points suggested by the authors. There are several things to be crucially provided.

Figure 1A showed that three kinds of exogenous genes were transformed to overexpress mHtt-Q74 in neuronal cells. The three plasmids were empty vector, Q23, and Q74. As described in "Materials and methods," the used plasmid DNA were GFP tagged construct, which can confirm the genes successfully transfected and expressed in cells. In this Figure, the GFP signals are needed to provide along with the bright field images. With the present images, it cannot validate whether Q74 was overexpressed in SK-N-MC. Only an increase of cell death in the cells transfected Q74 did not prove the overexpression of Q74.

Answer: Thanks for your important comments. As a result, we transfected Q23/Q74-Htt vectors without fluorescence marker in this revision. By reviewer’s suggestion, we have added RT-PCR results in Fig. 1B. This result showed that Q23 or Q74 is indeed overexpressed approximately 8 or 13 times higher in neuronal cells. We hope this can answer the reviewer’s question.

The construct of miR302 used in this study is needed in detail. According to "Materials and methods," the pLVX-AcGFP vector was modified for miR302 overexpression, and Figure 2A presented dsRed-RFP for confirming miR302 expression. It might that dsRed was fusioned to miR302 because Q74 was tagged with GFP.

Answer: Many thanks for reviewer’s important comments. To avoid the interference of dsRed on Rhodamine, we transfected miR-302s vector without fluorescence marker in this revision, and the result in Fig. 2A showed that miR-302 cluster is indeed overexpressed approximately 9 times higher than that in the mock group. Moreover, we also re-presented the results in Fig.3A, 3D, 4D, 5D to show an undisturbed state. Hope these corrections can make the results of this research more convincing.

Nuclear fragmentation was assessed just by DAPI staining, and it was quantified…….. I do not think it is an acceptable way. I recommend TUNEL staining for nuclear fragmentation.

Answer: By reviewer’s suggestion, we have added the result of TUNEL staining in Fig. 2E. This result demonstrated that mHtt-Q74-induced TUNEL signal could be reduced by upregulation of miR-302, suggesting overexpression of miR-302 indeed reduces apoptosis from mHtt-induced neurotoxicity.

There are needed to explain fluorescence images in Figure 3A, Figure 3D, Figure 4D, and Figure 5A. Q74 was tagged with GFP, and miR302 was tagged with dsRed. How could each fluorescence detect except tagging fluorescence proteins in Q74 and miR302 overexpressed neuronal cells? Another construct of Q74 or miR302 is used for Figure 3A, Figure 3D, Figure 4D, and Figure 5A? If it was, how could determine to overexpress Q74 or miR303 in the transfected cells?

Figure 5A showed the stained Htt with rhodamine to evaluate Htt aggregates. Why? Q74 was already tagged with GFP, and the GFP signals were shown as puncta in the nucleus of Q74 overexpressed cells. In addition, if the authors would suggest miR302 might work on Q74 overexpressing cells, co-localization of Q74 and miR302 in the same neuronal cells was displayed at least (/primarily).

Answer: Thank you very much for your question. The Em of Rhodamine is 565nm, and the Em of dsRed is 583nm (the green fluorescence also faces similar problems). Their wavelengths are so close that it is indeed may cause confusion. However, compared to the results of Htt punta or other organelle staining (the signal is generally concentrated in a small area), the fluorescent signal of overexpression vector is evenly filled in all areas, so we speculate that the stronger signal of the former can still be highlighted. However, this may still interfere with the real signal. In order to solve this problem, in tis revision we re-transfected Q23/Q74-Htt and miR-302s vector without fluorescence marker, and re-presented the results in Fig.3A, 3D, 4D, 5D to show an undisturbed state. Hope these results can convince the reviewer to demonstrate the possible protective effects of miR-302. We must thank the reviewer for helping us find this critical problem, so that we have the opportunity to carefully re-perform some experiments to eliminate possible doubts.

For detecting or quantifying the level of miR302, TaqMan assay was general, and there are commercial probes for miR302. Is there any specific reason for measuring the LARP7 exon 9-10 spliced product?

Answer: In fact, the polycistronic miR-302 cluster contains a/b/c/d four subtypes, and since this cluster is located between exon 9 and 10 of the LARP gene, the process of splicing is essential for this miR-302 release in natural cell physiology. The TaqMan assay is indeed a common method for measuring miR-302 a/b/c/d, such as the mirVana qRT-PCR primer sets (Ambion) that we have used. However, the TaqMan method must measure the four subtypes because of the different primer pairs. However, our method can indirectly measure the release of miR-302 cluster. Compared with the TaqMan method, it can show the regulation effect of cells on miR-302 cluster when facing adversity. Therefore, we innovatively use this method in this study to get closer to the real situation of the HD cell.

Round 2

Reviewer 3 Report

The authors have addressed all of my concerns and revised the manuscript including appropriately updated figures. I, therefore, recommend the publication of the manuscript.